# An Investigation on a Plane-Based Dynamic Calibration Method for the Handheld LiDAR Scanner

**DOI:** 10.3390/s22010369

**Published:** 2022-01-04

**Authors:** Shih-Hong Chio

**Affiliations:** Department of Land Economics, Chengchi University, No. 64, Sec. 2, ZhiNan Rd., Wenshan District, Taipei City 11605, Taiwan; chio0119@nccu.edu.tw; Tel.: +886-2-2939-3091 (ext. 51657)

**Keywords:** calibration, handheld LiDAR scanner, Velodyne VLP-16, least-squares adjustment

## Abstract

A plane-based dynamic calibration method had been proposed by the previous study for the GeoSLAM ZEB Horizon handheld LiDAR scanner. Only one preliminary test was presented. Therefore, three datasets in a calibration field were collected in this study on different dates and at different times on the same date to investigate the efficiency of the proposed calibration approach and calibration results. The calibration results for these three datasets showed that all average residuals were closer to 0, and all *a* posterior unit weight standard deviations of the adjustment were also significantly reduced after calibration. Moreover, the RMSE (root mean square error) of the check planes was improved by about an average of 32.61%, 28.44%, and 14.7%, respectively, for the three datasets. The improvement was highly correlated with the quality of the calibration data. The RMSE differences of all check planes using calibration data collected on different dates and at different times on the same date for calibration was about 1–2 cm and less than 1 mm, respectively. There was no difference in the calibration results, demonstrating the efficiency of the proposed calibration approach and the calibration results during the two different dates.

## 1. Introduction

The GeoSLAM ZEB Horizon GeoSLAM ZEB Horizon (GeoSLAM Ltd., Nottingham, UK) LiDAR scanner is a form of handheld mobile mapping system (MMS) and has been applied on many occasions, due to its compact size, cost-effectiveness, and high performance [1]. Due to its use of the simultaneous localization and mapping (SLAM) algorithm [1,2,3] with IMU (inertial measurement unit) data for positioning without using GNSS data [4], it can avoid environmental limits and be used in narrow and winding alleys, indoors, and in other areas where GNSS signals cannot be received. Compared to the total station and terrestrial LiDAR scanner, it also demonstrates high performance in collecting terrain data in general areas, such as in non-narrow alleys. Hence, it has been employed in different fields (e.g., cultural asset preservation in ancient cities [5], forest investigations [4,6,7], mine monitoring [1,8], disaster site reconstruction [9], tunnel surveying [10], topographic surveying [11], and the mapping of building interior structures [5,12]).

The Velodyne VLP-16 multibeam LiDAR sensor (Velodyne Lidar Inc., San Jose, CA, USA) is embedded in the GeoSLAM ZEB Horizon LiDAR scanner as the data collector. The Velodyne VLP-16 sensor comprises 16 individual laser transmitter pairs, which are individually aimed in 2° increments over the 30° field of view of the laser scanner (Figure 1).

The performance and calibration of the VLP-16, as one of the most popular multibeam spinning LiDAR sensors currently available on the market, have been studied and reported in the literature. The studies indicate that the calibration of the manufacturer’s Velodyne multibeam LiDAR sensor is not quite complete. After laboratory calibration, the accuracy of the point cloud could be further improved. One of the prerequisites for accuracy improvement of point clouds collected by the Velodyne multibeam LiDAR sensor is to eliminate the system error of the multibeam LiDAR sensors by calibration [15,16,17].

The calibration methods of the multibeam LiDAR sensors or scanners can be classified based on the method of obtaining the calibration data. One is called static calibration; the other is called dynamic calibration. In order to reduce the influence of the error caused by the incident angle and intensity value, and to obtain various ranging measurements and angle observation data, static calibration must place the sensor or scanner for calibration at different stations for data collection. For example, Glennie et al. [16] performed static calibration for the VLP-16 with planar features and studied its temporal stability. It was reported that the accuracy had increased by about 20% after calibration in a single collection. Chan et al. [13] considered the range scale factor of the range measurement in the static calibration of the VLP-16. The results showed that both the rangefinder offset and the range scale factor were considered; the RMSE of the check plane could be reduced by 30%. Meanwhile, the results were better than those with only either the rangefinder offset or the range scale factor considered. In summary, these studies demonstrated that self-calibration can further reduce the systematic error of the VLP-16. Since the range accuracy of the VLP-16 contributes significantly to the final data quality, it has been intensively studied with self-calibration experiments for the GeoSLAM ZEB Horizon.

Glennie [17] and Nouira et al. [18] installed the multibeam LiDAR sensor on a vehicle. They used the dynamically collected point cloud data for calibration, that is a dynamic calibration or kinematic calibration, eliminating the scanning demand on multiple stations to obtain point cloud data with different ranging measurements and improving the calibration efficiency.

The past study showed that the system error parameters could be more steadily calibrated if the calibration data contained uniformly distributed point cloud data at different ranging measurements. Although the accuracy of the specific ranging measurements would be significantly improved, the point cloud outside the specific ranging measurements would have a negative impact if the calibration data only contained certain specific ranging measurements [19].

In terms of system error parameter selection, system error parameters can generally be divided into physical correction parameters that have physically interpretable geometric meanings or empirical correction parameters that are obtained by statistical methods and can be explained empirically [20,21]. However, the empirical parameters require long-term statistical analysis, and the empirical parameters of different styles of scanners are not necessarily the same. In terms of physical parameters, Muhammad and Lacroix [19] supposed the rotating multibeam LiDAR consists of five system error parameters, which are the ranging system error ∆*r*, vertical angle error ∆α, horizontal angle error ∆β, vertical offset $V_{0}$, and the horizontal offset $H_{0}$ for each laser. Velodyne also sets five system error parameters for each laser of its multibeam LiDAR sensor HDL-64E S2 [15]. However, Glennie and Lichti [15], Glennie et al. [16] and Glennie [17] found that the vertical angle error ∆α and the horizontal angle error ∆β are highly correlated with the vertical offset $V_{0}$ and horizontal offset $H_{0}$ from the scanner frame origin for each laser in the static and dynamic calibration results. Glennie et al. [16] and Chan and Lichti [22] also indicated that the laser transmitter was a fixed component in the LiDAR scanner, precisely installed in the LiDAR scanner by the manufacturer. Thus, the vertical offset $V_{0}$, the horizontal offset $H_{0}$ for each laser, and the vertical angle error Δα are remarkably small compared to the ranging system error, which could be ignored in the system error parameters. Chan and Lichti [22] stated that the vertical offset $V_{0}$ can be regarded as the radial distance $\varepsilon_{j}$ between the scanning center, and the origin multiplied by $\sin\alpha_{j}$, and $\varepsilon_{j}$ can be regarded as part of the ranging system error ∆*r*. Therefore, the vertical offset $V_{0}$ can be absorbed by Δr and can be removed from the system error parameters.

In the calibration results of Chan et al. [13], it was found, on the contrary, that in the VLP-16 ranging system error parameter, if only a single parameter of the range scale factor or the rangefinder offset was included in each beam, it would have a negative impact on the RMSE of the check planes after calibration. The error of the ranging system must be included in the range scale factor and the rangefinder offset at the same time in each beam to stably reduce the RMSE of the check planes. However, the correlation coefficient would probably be too high between the range scale factor and the rangefinder offset while the scanning distance is less than 50 m. If the range scale factor and the rangefinder offset are considered simultaneously, the long-ranging measurement calibration data must be appropriately increased [15].

Considering the scanning point density of the VLP-16, it is more feasible to use geometric features instead of calibration targets. Plane features are the most easily obtained features of an indoor environment. The GeoSLAM ZEB Horizon can directly use dynamic scanning to collect data for calibration, avoiding the scanning problem of multiple stations while saving time [17]. Meanwhile, dynamic scanning can obtain more various ranging measurements and increase the reliability of calibration. The autorotation of the multibeam LiDAR scanner can avoid the limitation where the multibeam LiDAR scanner must be scanned at different tilt angles to obtain calibration data from different angles. The horizontal offset $H_{0}$ and the vertical angle error ∆α are less significant than other parameters. On the other hand, the horizontal angle error ∆β only has a significant impact on long-distance observations [17]. Moreover, the GeoSLAM ZEB Horizon is usually for short-distance scanning, and the angle error has a low impact (e.g., data collection in narrow and winding alleys for building information). Therefore, the horizontal angle error Δβ could be ignored, and the system error calibration is only performed for the range scale factor and the rangefinder offset for the GeoSLAM ZEB Horizon. However, the literature has focused on the multibeam LiDAR scanner. Each laser transmitter in the scanner has a different set of system error parameters [13,16]. The VLP-16 has 16 laser transmitters; that is, there are 16 sets of system error parameters. Moreover, the GeoSLAM ZEB Horizon installs the Velodyne VLP-16 in a rotating mechanism, which rotates while scanning. Therefore, it is difficult to convert the horizontal angle error in the horizontal state. Additionally, the GeoSLAM ZEB Horizon cannot output the laser transmitter parameters in the scanning results. In particular, the laser transmitter of each point data cannot be identified. Thus, only a set of system error parameters could be assumed for all data. Therefore, the ranging system error ∆*r* is described by the range scale factor (*S*) and the rangefinder offset (*C*).

Accordingly, a plane-based dynamic calibration method for the GeoSLAM ZEB Horizon was proposed by the previous study [23] to calculate the systematic errors, including the range scale factor (*S*) and the rangefinder offset (*C*), and an indoor environment with sufficient space was selected as the calibration field. The plane parameters were estimated from the higher precision ground-based LiDAR scanner for calibration planes and check planes. Calibration points on the surfaces of planar features were collected kinematically, extracted manually, and noise point clouds were removed by the RANSAC algorithm [24]. The proposed calibration method estimated the range scale factor (*S*) and the rangefinder offset (*C*) of the GeoSLAM ZEB Horizon simultaneously with the coordinate transformation parameters by transforming the corrected handheld point clouds to lie on the surfaces of calibration planar features in order to minimize the sum of the square of the residuals. Although the results verified that the residuals could be reduced, and the check plane accuracy improved by an average of 41% by calibrated ranging system error correction, the calibration data with only short-ranging measurements were used. Furthermore, only one preliminary test was presented, and no more advanced study was investigated. Therefore, this study used calibration data with long-ranging measurements to further investigate the calibration results using calibration data collected on different dates and at different times on the same date.

## 2. Methodology

The study steps are described as follows. Firstly, an indoor calibration field was selected. Then, the points of this calibration field were collected by a ground-based LiDAR scanner, and the plane features in the calibration field were determined as the calibration planes and check planes. Subsequently, the points on the corresponding calibration planes were dynamically collected by a handheld LiDAR scanner, the GeoSLAM ZEB Horizon, on different dates and at different times on the same date. The points corresponding to the calibration planes and check planes were selected manually. Simultaneously, the blunder points were removed by the RANSAC algorithm. The plane-based dynamic calibration method for the GeoSLAM ZEB Horizon proposed by the previous study [23] was used to calculate the ranging systematic errors, including the range scale factor (*S*) and the rangefinder offset (*C*), by the least-squares adjustment method. The calibration results were investigated. The details for each step were described in the following sections.

### 2.1. Selection of the Calibration Field

The calibration field was full of plane features and large enough to collect point clouds with various ranging measurements to extract planes as reference data for calibration.

### 2.2. Acquisition and Extraction of Calibration Reference Data

#### 2.2.1. Ground-Based LiDAR Point Cloud Acquisition

In this study, a Faro ground-based LiDAR scanner was used to capture the point cloud of the calibration site in one station instead of multiple stations to avoid point cloud registration errors for accurate calibration reference data. The specification of the FARO Focus S350 ground-based LiDAR scanner (Faro Technologies Inc., Lake Mary, FL, USA) is tabulated in Table 1. The FARO Focus S350, a phase-based 3D laser scanner, has better accuracy than a 3D laser scanner based on a time of flight (TOF) scanning technique for collecting points in short distances under indoor conditions without interference.

#### 2.2.2. Extraction of Calibration Planes and Check Planes

This study used the plane features abundant in building interior environments as the calibration reference data, eliminating the need for a large number of construction procedures for the calibration targets. The plane equation of the plane feature *k* is shown in Equation (1), as follows.
(1)$$a_{k}x+b_{k}y+c_{k}z+d_{k}=0$$
where $\left(a_{k},b_{k},c_{k}\right)$: a unit normal vector of plane feature *k*.

The plane parameters ($a_{k},b_{k},c_{k},d_{k}$) were regarded as a priori parameters in the calibration process and served as the calibration reference data. As mentioned in Section 2.2.1, to improve the precision and efficiency of data acquisition, a FARO Focus S350 ground-based LiDAR scanner was operated to acquire point clouds of the calibration field, while the points located on the used plane features were manually extracted. The extracted plane points were used to determine the plane parameters by the least-squares method. Moreover, the extracted plane features were classified into the calibration planes and check planes. The calibration planes were used to determine the planar parameters for calibration, and the check planes were used to verify the calibration result.

### 2.3. Acquisition of Handheld LiDAR Point Cloud for Calibration Data

The GeoSLAM ZEB Horizon handheld LiDAR scanner (see Table 2 for technical specifications) was used to collect the point cloud data of the calibration field in a mobile manner to obtain as many point clouds on various plane features as possible for calibration data. According to the manual of the GeoSLAM ZEB Horizon, the scanning path must be close to the starting point of the trajectory in order to employ the SLAM algorithm [1,2,3] together with the IMU data to calculate a better scanning trajectory.

#### 2.3.1. Filtering and Subsampling of Point Clouds

According to the studies of Glennie and Lichti [15] and Glennie [17], if the Velodyne multibeam LiDAR sensor has an excessively large incident angle to the object surface during scanning, it is likely to cause a significant increase in point cloud noise due to the decrease in reflection intensity. Although the GeoSLAM ZEB Horizon scans the data with the VLP-16 in a rotating manner, it might also decrease the accuracy of the point cloud due to the large incident angle. In order to reduce the influence of errors caused by factors other than the ranging measurement of the handheld LiDAR scanner, GeoSLAM Hub software was used to output the normal information of each point and the SLAM calculation quality (SLAM condition) to calculate the incident angle λ of each point, and to obtain the quality index of the trajectory calculation to filter the point clouds. Points with large incidents and bad SLAM conditions were filtered out.

After the point cloud was filtered, the point clouds located on the calibration planes and check planes were manually extracted for the calibration data. In order to avoid the large difference in the point numbers on different planes and the excessive concentration in certain ranging measurements to affect the determination of the ranging system error parameters— including the range scale factor (*S*) and the rangefinder offset (*C*)—the same number of point clouds was randomly subsampled on each plane so that the calibration point data could be evenly collected in the various ranging measurements.

#### 2.3.2. Blunder Point Filtering Using the RANSAC Algorithm

In order to avoid including error or noisy points in the determination of the range scale factor (*S*) and the rangefinder offset (*C*), the blunder points of the subsampled points were removed using the RANSAC algorithm [24]. RANSAC is an algorithm for estimating a specific mathematical model from a sample containing gross errors. A fixed amount of data is randomly sampled from the sample to calculate a mathematical model that matches the sampled data. The remaining data after sampling are substituted into this mathematical model, and the residual is calculated. If the residual error is less than the given threshold, the data are regarded as the inner group conforming to the mathematical model. If the residual error is greater than the given threshold, the data are considered a gross error or blunder. The above steps are repeated, and the largest number of inner groups conforming to the mathematical model is regarded as the best model parameter in classifying and locating gross error data.

### 2.4. Mathematical Model for Calibration

Based on the previous study [23], the ranging system error *∆r* was only discussed and described by the range scale factor (*S*) and the rangefinder offset (*C*). The ranging measurement *r* after the correction is expressed in Equation (2). It was considered the adjustment system’s additional parameters (APs) and solved in the least-squares adjustment.
(2)$$r_{icorrect}=r_{i}*S+C$$
where $r_{i}$: original ranging measurement;$\;r_{icorrect}$: ranging measurement after ranging error correction.

#### 2.4.1. Scanning Center Determination of Each Point

The GeoSLAM ZEB Horizon cannot output the original ranging measurements. In order to obtain each ranging measurement $r_{i}$, called the pseudoranging measurement, the laser scanning center coordinates corresponding to each point should be obtained from the trajectory data. The trajectory data could be output by the GeoSLAM Hub software. The recording frequency in the trajectory data was 0.01 seconds. Therefore, the corresponding laser scanning center coordinates ($x_{ic},y_{ic},z_{ic}$) were determined by the linear interpolation formula, as follows:(3)$$x_{ic}=x_{c0}+\left(t_{i}-t_{0}\right)\frac{x_{c1}-x_{c0}}{t_{1}-t_{0}};\;y_{ic}=y_{c0}+\left(t_{i}-t_{0}\right)\frac{y_{c1}-y_{c0}}{t_{1}-t_{0}}\;z_{ic}=z_{c0}+\left(t_{i}-t_{0}\right)\frac{z_{c1}-z_{c0}}{t_{1}-t_{0}}$$
where:
$\left(x_{ic},\;y_{ic},\;z_{ic}\right)$: the corresponding laser center coordinates of point *i*.$t_{i}$: the scanning time of point *i*.$\left(x_{c0},\;y_{c0},\;z_{c0}\right)$: the trajectory coordinates that the scanning time of point *i* is less than $t_{i}$ but is closest to $t_{i}$.$\left(x_{c1},\;y_{c1},\;z_{c1}\right)$: the trajectory coordinates that the scanning time of point *i* is large than $t_{i}$ but is closest to $t_{i}$.$t_{0}$: the time of the calculated trajectory that is less than the scanning time $t_{i}$ for point *i* and is closest to $t_{i}$.$t_{1}$: the time of the calculated trajectory that is larger than the scanning time $t_{i}$ for point *i* and is closest to $t_{i}$.


By using the laser center coordinates of point *i*, the space vector ($\Delta x_{i},\;\Delta y_{i},\;\Delta z_{i}$) of point *i* was calculated by Equation (4), as follows.
(4)$$[\begin{array}{c} \Delta x_{i}\\ \Delta y_{i}\\ \Delta z_{i} \end{array}]=[\begin{array}{c} x_{i}\\ y_{i}\\ z_{i} \end{array}]-[\begin{array}{c} x_{ic}\\ y_{ic}\\ z_{ic} \end{array}]$$

The calculated pseudoranging measurement $r_{i}$, the horizontal angle $\alpha_{i}$, and the vertical angle $\beta_{i}$ should be calculated according to the following Equations (5)–(7) for subsequent derivation.
(5)$$r_{i}=\sqrt{\Delta x_{i}{}^{2}+\Delta y_{i}{}^{2}+\Delta z_{i}{}^{2}}$$
(6)$$\alpha_{i}=\tan^{-1}\frac{\Delta z_{i}}{\sqrt{\Delta x_{i}{}^{2}+\Delta y_{i}{}^{2}}}$$
(7)$$\beta_{i}=\tan^{-1}\frac{\Delta y_{i}}{\Delta x_{i}}$$

Through the pseudoranging measurement $r_{i}$, the horizontal angle $\alpha_{i}$, the vertical angle $\beta_{i}$, and the corresponding laser center coordinates ($x_{ic},y_{ic},z_{ic}$), the coordinates of point *i* could be reconstructed as shown in Equation (8). The pseudoranging measurement $r_{i}$ was regarded as the ranging measurement measured by the handheld LiDAR scanner. To substitute the calibration ranging error APs (*S* for the ranging scale factor and *C* for the rangefinder offset) into Equation (8), the coordinates of point *i* could be determined by Equation (9), as follows:(8)$$[\begin{array}{c} x_{i}\\ y_{i}\\ z_{i} \end{array}]=[\begin{array}{c} r_{i}*\cos\alpha_{i}*\sin\beta_{i}\\ r_{i}*\cos\alpha_{i}*\cos\beta_{i}\\ r_{i}*\sin\alpha_{i} \end{array}]+[\begin{array}{c} x_{ic}\\ y_{ic}\\ z_{ic} \end{array}]$$
(9)$$[\begin{array}{c} x_{i}\\ y_{i}\\ z_{i} \end{array}]=[\begin{array}{c} \left(r_{i}*S+C\right)*\cos\alpha_{i}*\sin\beta_{i}\\ \left(r_{i}*S+C\right)*\cos\alpha_{i}*\cos\beta_{i}\\ \left(r_{i}*S+C\right)*\sin\alpha_{i} \end{array}]+[\begin{array}{c} x_{ic}\\ y_{ic}\\ z_{ic} \end{array}]$$

#### 2.4.2. Mathematical Model for Calibration

The data obtained by the ground-based LiDAR scanner and the handheld LiDAR scanner were respectively located in the ground-based LiDAR coordinate system and the handheld LiDAR coordinate system. Only when the LiDAR point cloud was converted to the ground-based LiDAR coordinate system could the plane parameters obtained by the ground-based LiDAR scanner be used as the calibration reference data to solve the ranging APs. Therefore, the six rigid-body conversion parameters (three translation and three rotation parameters) were considered as the unknowns and added to the adjustment equation for simultaneous determination. Equation (10), as shown below, describes the conversion of the handheld LiDAR point ($x_{i},\;y_{i},\;z_{i}$) after the correction of the ranging system error to the ground-based LiDAR coordinate system through the six rigid-body conversion parameters.
(10)$$[\begin{array}{c} X_{i}\\ Y_{i}\\ Z_{i} \end{array}]=R\left(\kappa \right)R\left(\varphi \right)R\left(\omega \right)*[\begin{array}{c} x_{i}\\ y_{i}\\ z_{i} \end{array}]+[\begin{array}{c} X_{t}\\ Y_{t}\\ Z_{t} \end{array}]=R*[\begin{array}{c} x_{i}\\ y_{i}\\ z_{i} \end{array}]+[\begin{array}{c} X_{t}\\ Y_{t}\\ Z_{t} \end{array}]$$
where *R*: rotational transformation matrix; ($X_{t}$,$Y_{t}$,$Z_{t}$): translation vector.

Equation (10) is the main equation of the plane-based dynamic calibration method developed by a previous study [23]. It also means that point *i* should be located on the corresponding calibration plane fitting with the point cloud from the ground-based LiDAR scanner, after being corrected by the range scale factor and rangefinder offset and conversion. Due to the random error, the converted point could not be located on the calibration plane. Thus, to minimize the sum of distance squares from the converted points to their corresponding calibration planes, a mathematical model of adjustment was developed to determine the ranging APs. The mathematical model included the functional and the stochastic models. The observation equations were regarded as the identical weight. Equation (11) shows the functional model for the least-squares adjustment. The calibration plane parameters ($a_{k},\;b_{k},\;c_{k}$) used in this study were unit vectors, so the above-mentioned sum of distance squares from the converted points to their corresponding calibration planes could be regarded as the squares sum of correction $v_{i}$ (i.e., residuals). The pseudoranging observation equation is shown in Equation (11). Each handheld LiDAR point establish a pseudo-observation equation. All of the pseudo-observation equations were used to simultaneously solve the ranging APs and the six rigid-body conversion parameters according to the least-squares principle.
(11)$$F_{n}=\left[\begin{array}{ccc} a_{k} & b_{k} & c_{k} \end{array}\right][\begin{array}{c} X_{i}\\ Y_{i}\\ Z_{i} \end{array}]+d_{k}=0+v_{i}$$
where $a_{k,}\;b_{k,}\;c_{k,}\;d_{k}$: plane parameters of plane *k*; ($X_{i},\;Y_{i},\;Z_{i}$): the coordinates of point *i* on the ground-based LiDAR system after rigid-body conversion.

The unknowns of this adjustment were 2 ranging APs (*S* and *C*) and 6 coordinate transformation parameters. Since Equation (11) is a nonlinear equation, it should be linearized by Taylor expansion to establish the indirect observation adjustment matrix form (see Equation (12) required by the least-squares method). The pseudo-observation equations were regarded as equal weights, and the corrections of the initial values of the unknowns were determined by Equation (13). Then, the corrections were added to the initial value before iterations to reorganize the indirect observation adjustment matrix (see Equation (12)). During iterations, the threshold was set as the ratio of the *a* posterior variance change less than 0.000001 (see Equation (14)). The ranging APs and the six rigid-body conversion parameters could be solved until the *a* posterior variance change ratio converged.
(12)$$JX=K+V\;J=\left[\begin{array}{cc} \begin{array}{cc} \frac{\partial F_{1}}{\partial S} & \frac{\partial F_{1}}{\partial C}\\ \frac{\partial F_{2}}{\partial S} & \frac{\partial F_{2}}{\partial C} \end{array} & \begin{array}{cc} \cdots & \frac{\partial F_{1}}{\partial Z_{t}}\\ \cdots & \frac{\partial F_{2}}{\partial Z_{t}} \end{array}\\ \begin{array}{cc} \vdots & \vdots \\ \frac{\partial F_{n}}{\partial S} & \frac{\partial F_{n}}{\partial C} \end{array} & \begin{array}{cc} \vdots & \vdots \\ \cdots & \frac{\partial F_{n}}{\partial Z_{t}} \end{array} \end{array}\right],X=[\begin{array}{c} dS\\ dC\\ \begin{array}{c} d\varphi \\ d\omega \\ \begin{array}{c} d\kappa \\ dX_{t}\\ \begin{array}{c} dY_{t}\\ dZ_{t} \end{array} \end{array} \end{array} \end{array}],K=\left[\begin{array}{c} -F_{10}\\ -F_{20}\\ \begin{array}{c} \vdots \\ -F_{n0} \end{array} \end{array}\right]$$
where ***J***: Jacobian matrix; ***X***: correction vector of the initial value of the unknowns; ***K***: the difference vector between 0 and the value of substituting the initial value into pseudo-observation equations; ***V***: the residual vector of the pseudo-observation equations; *n*: the number of pseudo-observation equations.
(13)$$X={\left(J^{T}J\right)}^{-1}\left(J^{T}K\right)$$
(14)$$Ratio=\left|\frac{\sigma_{0,i}{}^{2}-\sigma_{0,i-1}{}^{2}}{\sigma_{0,i-1}{}^{2}}\right|\;$$
where $\sigma_{0,i-1}{}^{2}$: Variance in the (*i*−1)th iteration; $\sigma_{0,i}{}^{2}$: Variance in the *i*th iteration.

### 2.5. Result Analysis

The calibration results using calibration data collected on two different dates and at two different times on the same date were discussed. The result analysis in this study includes residuals analysis, the RMSE verification of check planes, the analysis of the correlation matrix of the unknowns, and the result analysis of the calibrated range scale factor (*S*) and the rangefinder offset (*C*) parameters.

#### 2.5.1. Residuals Analysis

The influences on the residual distribution, the average value of the residuals, and the *a* posterior unit weight standard deviation, before and after the ranging system error correction, were used to verify whether the residual error distribution had the implicit system error or not, and whether the average value of the residual error and the standard deviation had decreased or not in order to evaluate if the calibrated ranging system error parameters could correct the system error and improve the accuracy of the handheld LiDAR points.

#### 2.5.2. Verification by the RMSE of Check Planes

The RMSE of calibration data from corrected and uncorrected handheld point clouds to each corresponding check plane, before and after the adjustment, was calculated to evaluate calibration results. The improvement ratio for each check plane was calculated by Equation (15) to verify the efficiency of the system error correction, as follows.
(15)$$ratio=\frac{\left(RMSE_{with\;APs}-RMSE_{without\;APs}\right)}{RMSE_{without\;APs}}$$
where:
$RMSE_{with\;APs}$: The RMSE, calculated by adding the ranging additional parameters (APs) to the adjustment.$RMSE_{without\;APs}$: The RMSE, calculated by not adding the ranging additional parameters (APs) to the adjustment.


#### 2.5.3. Analysis of Correlation Matrix of the Unknowns

Through the correlation coefficient matrix, the quality and robustness of the calibration results [13] can be verified. It can also check whether there is a high correlation between the error parameters of the ranging system and the conversion parameters. High correlation means that the calibration method or calibration data are insufficient to solve the calibration parameters well. The correlation coefficients after the adjustment were shown and discussed in this study.

#### 2.5.4. Analysis of Ranging Systematic Error Parameters

In this study, two systematic errors were estimated: *S* for the range scale factor and *C* for the rangefinder offset. The influence for a certain ranging measurement (e.g., 20 m, 30 m, and 40 m) was investigated.

## 3. Experimental Description

### 3.1. Selection of the Calibration Field

The size of the calibration site was about 35 m by 27 m by 3 m, located on the underground parking lot of the Research and Innovation–Incubation Center at National Chengchi University in Taiwan. The calibration site provided a large variety of planar features with different ranges for calibration, as shown in Figure 2.

### 3.2. Acquisition and Extraction of Calibration Reference Data

As mentioned in Section 2.2.1, a Faro ground-based LiDAR scanner was used to capture the point cloud of the calibration site in one station instead of multiple stations to avoid point cloud registration errors. The collected point cloud data are shown in Figure 3. Seventeen plane features labeled A to Q were manually extracted from the ground-based LiDAR point cloud for calibration reference data. The location of each plane is shown in Figure 4. The size of each plane was about 0.7 m by 1.2 m, and its planar parameter was determined by the least-squares method. The parameters and fitting RMSEs are shown in Table 3. All plane-fitting RMSEs were not greater than 0.001 m, indicating that the point cloud data of FARO Focus S350 was of a certain accuracy and reliability as the calibration planes and the check planes.

In order to solve the ranging APs simultaneously, together with the six rigid-body conversion parameters, three planes with orthogonal normal vectors must be included in the plane selection [13]. The horizontal and vertical planes could be identified from the DIP in Table 3. Table 3 shows the planes, including 4 horizontal (DIP 0°) and 13 vertical (DIP 89°) planes from the DIPs. The triple pair (a, b, c) indicated the unit normal vector in Table 3.

### 3.3. Acquisition of Handheld LiDAR Point Cloud for Calibration Data

A plane-based dynamic calibration developed in the previous study [23] and mentioned in more detailed in Section 2.4 was performed to investigate the calibration results. Dynamic or kinematic calibration means capturing the point cloud data in a mobile manner for calibration. Compared to static calibration, dynamic calibration can obtain richer points with various ranging measurements, and there is no need to place the handheld LiDAR scanner in multiple stations to capture data separately, saving a lot of time [17]. For the tests in this study, three calibration datasets of the point cloud in the calibration field were obtained by the GeoSLAM ZEB Horizon. The relevant collected information was tabulated, as shown in Table 4. The scanning time for each dataset was approximately 85 sec, and the number of point clouds for each dataset was about 150,000,000 points. Figure 5, Figure 6 and Figure 7 show the collected datasets, and the colors in Figure 5a, Figure 6a and Figure 7a are determined based on the SLAM quality (SLAM condition). The best quality is blue (R = 0; G = 0; B = 255); the closer the datasets are to red, the worse the quality. In an indoor environment with rich features, the SLAM quality was obviously stable, and most of the point clouds were dark blue, indicating no significant problem in the SLAM solution or failure of the SLAM solution. The color of the point cloud in Figure 5b, Figure 6b and Figure 7b was given according to the scanning time, and the color of the point cloud gradually changed from red to blue according to the sequence of time; red was the scanning time of the beginning, and blue was the scanning time of the end. The scanning trajectory is shown in Figure 8, Figure 9 and Figure 10. The scanning path was planned as a walk around the center of the parking lot and close to the starting point at a normal walking speed to ensure that complete calibration data were scanned. The color-setting method of the trajectory was the same as in Figure 5b, Figure 6b and Figure 7b. It could be found that the starting point represented by red and the scanning end point represented by blue were close, which met the scanning requirements. Figure 8, Figure 9 and Figure 10 demonstaate the point cloud colored by scanning time and the scanning trajectory of dataset DATA1, DATA2, and DATA3.

### 3.4. Filtering, Subsampling, and Blunder Removing of Point Clouds

In this study, the filtering conditions of the handheld LiDAR point cloud were performed by the SLAM quality and the incident angle of the point. In order to remove the influence caused by the poor SLAM quality, the threshold value was set to R = 0, G = 0, and B = 255; that is, the points collected by the best SLAM solution were used as calibration data. Based on the studies of Glennie and Lichti [15] and Glennie [17], the thresholds of the incidence angle for the effectiveness of filtering with the incident angle were 60° and 70°, respectively. However, the point cloud filtered with 60° and the plane features in the horizontal direction were relatively insufficient for DATA1. Considering that a certain number of plane features in the horizontal direction must be extracted, this study used 70° as the filter threshold for the incident angle.

In order to evaluate the effectiveness in filtering poor-quality point clouds by the thresholds, a point cloud on one plane was taken from the dataset DATA1 for analysis to discuss whether the RMSE of the plane fitting after filtering was reduced or not. The selected plane location and its corresponding point cloud data are shown in Figure 11, and the analysis results are shown in Table 5. After filtering using the SLAM quality or incident angle separately, the plane fitting RMSE was reduced from 0.0110 m to 0.0108 m. If both the SLAM quality and incident angle were used for filtering, the plane fitting RMSE was further reduced to 0.0106 m. The results showed that the point cloud’s plane fitting RMSE by the three filtering conditions was better than the original point cloud used for plane fitting. It could be said that the used conditions and thresholds could retain the point cloud with better observation conditions. Therefore, this study used these combined conditions for filtering. Figure 12, Figure 13 and Figure 14 show the point cloud data after filtering for the datasets DATA1, DATA2, and DATA3. After the point cloud filtering, the number of point clouds for the three datasets is 1,401,803, 1,729,824, and 1,804,708 points. Meanwhile, the filtering ratio is about 91%, 88%, and 88% for the datasets DATA1, DATA2, and DATA3, respectively. The remaining good-quality point cloud data were employed as calibration data for calibration adjustment.

The handheld LiDAR point cloud corresponding to each plane (see Figure 4) was selected manually, and the point number contained on each plane and the ranging measurement between each point relative to its corresponding laser center were calculated. The coordinates of each laser center can be seen in Equation (3). The statistical results of calculated pseudoranging measurements for each plane of dataset DATA1 are shown in Table 6. Further, the statistics for pseudoranging measurements were divided into minimum, maximum, median, and average to evaluate the pseudoranging measurements provided by the points in each plane for calibration adjustment calculation. Table 6 shows that planes B, I, and J have relatively fewer points, due to the longer scanning distances. Plane E is blocked by a wall, while the point number was also relatively insufficient.

In order to increase the calculation efficiency and to allow the calibration calculation to include uniform and various pseudoranging measurements, the points on each plane were randomly subsampled to select the same number of point clouds as the calibration calculation data, except for planes B, E, I, and J, which were of a lower number of points. The least number of points on the other planes is plane D, and the point number was 613. Therefore, this study randomly subsampled 600 points for the remaining planes and evaluated whether there was a significant difference in the calculated pseudoranging measurements before and after subsampling. After subsampling, there were some blunder points. In this study, the RANSAC algorithm was used to remove the gross errors of the subsampled points in each plane for the three datasets. According to the point cloud precision announced by the original GeoSLAM ZEB Horizon manufacturer, the allowable error threshold was set to 0.03 m. The point numbers and the statistics of calculated pseudoranging measurements in each plane after subsampling and blunder removal by the RANSAC algorithm for the dataset DATA1 are shown in Table 7.

The box diagram of the calculated pseudoranging measurements before and after subsampling and blunder removal by the RANSAC algorithm for the dataset DATA1 is shown in Figure 15. Table 7 and Figure 15 indicate no significant difference between the minimum, maximum, median, and average values of the calculated pseudoranging measurements in each plane before and after subsampling, and blunder removal by the RANSAC algorithm. The difference is less than 1 meter, indicating that the same rich calculated pseudoranging measurements could be retained after subsampling. Therefore, this study used 600 points as the number of subsampling points, taking into account the richness of the calibration data and improving the calculation efficiency. Taking plane K as an example, the subsampling results are shown in Figure 16. Table 8 and Table 9 show the point numbers and the statistics of calculated pseudoranging measurements in each plane for the datasets DATA2 and DATA3 after subsampling and blunder removal. Figure 17 illustrates the box diagram of calculated pseudoranging measurements in each plane after subsampling and blunder removal for the datasets DATA1, DATA2, and DATA3. Figure 17 indicates again no significant difference between the minimum, maximum, median, and average values of the calculated pseudoranging measurements in each plane, before and after subsampling and blunder removal by the RANSAC algorithm for the three datasets.

### 3.5. Residuals Analysis

A plane-based dynamic calibration proposed by a previous study [24] and mentioned in Section 2.4 was performed. The location distribution of the calibration planes in this study is shown in Figure 18. The selection of the calibration planes should enclose the entire calibration field as much as possible and be evenly distributed. At the same time, in addition to the vertical planes, the calibration planes must also include the horizontal planes (e.g., planes A, O, Q) [13] to simultaneously solve the rigid-body conversion parameters during the calibration adjustment calculation. In particular, although plane E seemed to be closer to the periphery of the calibration field than plane H, plane H was of a richer scanning distance than plane E from the maximum and minimum values in Table 7, Table 8 and Table 9. Thus, plane H instead of plane E was selected as the calibration plane. The following Sections investigate the calibration results.

#### 3.5.1. Residuals Analysis

Figure 19, Figure 20 and Figure 21 are the residual distribution plots. Meanwhile, Figure 22, Figure 23 and Figure 24 show the residual scatter plots using the datasets DATA1, DATA2, and DATA3 after adjustment, with or without determination of the ranging system error, respectively. In particular, Figure 19a, Figure 20a and Figure 21a are the adjustment results’ residual distribution or residual scatter plots, by incorporating the ranging system error into the adjustment system as additional parameters (referred to with APs) together with the six rigid-body conversion parameters were regarded as the unknowns for determination simultaneously. Meanwhile, Figure 19b, Figure 20b and Figure 21b are the adjustment results’ residual distribution or residual scatter plots, without adding the ranging system error as the additional parameters (referred to without APs), and only the six rigid-body conversion parameters were retained as the unknowns for determination.

The residual dispersion in Figure 19a, Figure 20a and Figure 21a for the three datasets was similar. Comparing Figure 19a, Figure 20a and Figure 21a to Figure 19b, Figure 20b and Figure 21b, the residual value distributions were more concentrated at 0 and were more in line with the normal distribution curve. After further analysis by the residual scatter diagram, it could be seen from Figure 22, Figure 23 and Figure 24 that the residuals in Figure 22a, Figure 23a and Figure 24a were stably and evenly dispersed within ±0.03 m, conforming to the 3 cm precision of point cloud announced by the manufacturer. The residuals in Figure 22b, Figure 23b and Figure 24b were affected by more significant systematic errors, which caused the residual dispersion to present unstable undulation. Table 10 tabulates the average residuals and the *a* posterior unit weight standard deviation for the three datasets, using least-squares adjustment, with and without ranging APs. For the three datasets, after adding the ranging APs, the average residuals were closer to 0, and the *a* posterior unit weight standard deviation became smaller, both of which were improved compared to those without adding the ranging APs in the adjustment. Based on the above deduction, adding the ranging APs in this study could eliminate most of the GeoSLAM ZEB Horizon’s ranging system errors.

#### 3.5.2. Verification by the RMSE of Check Planes

By evaluating the calibration results, the RMSE of each check plane was calculated for the three datasets using least-squares adjustment, with and without ranging APs. Table 11, Table 12 and Table 13 show the RMSE results of each check plane for the three datasets. Table 11 shows the RMSE results of the check planes using the dataset DATA1. Among them, the RMSE of all check planes improved after correcting the ranging system error (see the RMSE line chart of each check plane in Figure 25). With up to 72.12% in plane F, an increase of about 2.4 cm and an improvement of 1.6 cm in plane E were reached. The overall average improvement was 32.61%, demonstrating that the proposed calibration approach could effectively improve the overall point cloud accuracy of the GeoSLAM ZEB Horizon.

Table 12 and Table 13 show the RMSE results of the check planes, using the datasets DATA2 and DATA3 at different times on the same day. For the dataset DATA2, the RMSE of check plane P was not improved. However, the difference was only 0.0001 m. The RMSEs of the remaining 7 planes were improved after correcting the ranging system error (see the RMSE line chart of each check plane in Figure 26). With up to 61.08% in plane F, an increase of about 1.8 cm and an improvement of 1.4 cm in plane E were reached. The overall average improvement was 28.44%.

For the dataset DATA3, the RMSE of check plane C was not improved. However, the difference was only 0.0003 m. The RMSEs of the remaining 7 planes were improved after correcting the ranging system error (see the RMSE line chart of each check plane in Figure 27). With up to 54.77% in plane E, an increase of about 1.4 cm and an improvement of 0.8 cm in plane F were reached. The overall average improvement was 14.70%. The improvement of dataset DATA3 was less than that of the dataset DATA2. Meanwhile, the quality of the point cloud in dataset DATA3 was better than that of the dataset DATA2 because the mean ${\mathrm{RMSE}}_{\mathrm{without}\;\mathrm{APs}}$ of dataset DATA3 (0.0160 m) was less than the mean ${\mathrm{RMSE}}_{\mathrm{without}\;\mathrm{APs}}$ of dataset DATA2 (0.0194 m). The RMSE in plane I was still larger than about 2 cm among the three datasets (see Table 11, Table 12 and Table 13 and Figure 25, Figure 26 and Figure 27) after the ranging system error correction. This finding can be explained in that there were only 58, 62, and 57 points on the said check plane after blunder detection by the RANSAC algorithm (see Table 7, Table 8 and Table 9). The calibration results for the three datasets on different dates and at different times showed that the check plane accuracy improved by an average of 32.61%, 28.44%, and 14.7%, respectively. The improvement is highly correlated with the quality of the calibration data. From the mean ${\mathrm{RMSE}}_{\mathrm{without}\;\mathrm{APs}}$ shown in Table 11, Table 12 and Table 13, the result of dataset DATA3 was more accurate than that of the datasets DATA1 and DATA2. After calibration, the mean ${\mathrm{RMSE}}_{\mathrm{with}\;\mathrm{APs}}$ was about 1 cm, which is better than the approximately 2 cm mean ${\mathrm{RMSE}}_{\mathrm{without}\;\mathrm{APs}}$ before calibration. The improvement of the check planes’ RMSE reclaimed that the proposed calibration approach could effectively improve the overall point cloud accuracy of the GeoSLAM ZEB Horizon.

Table 14 shows the comparison of the RMSE difference of each check plane for the three datasets, after calibration on different dates and at different times on the same date. Using data collected on different dates for calibration (the datasets DATA2 and DATA1, DATA3 and DATA1), the difference in the mean RMSE difference of the two sets of results was 0.0016 m. Using the same date but different times (the datasets DATA3 and DATA2) to calibrate, the difference in the mean of RMSE difference was 0.0001 m. Those were not significant differences statistically, thereby showing the stability of the calibration results in this test.

#### 3.5.3. The Analysis of Correlation Matrix of the Unknowns

Table 15, Table 16 and Table 17 are the matrices of correlation coefficients of the unknowns of the least-squares calibration solution for the three datasets. The correlation coefficients between the ranging APs and the coordinate conversion parameters were maintained at a low correlation, and the absolute values of the correlation coefficients were mostly less than 0.7. Only the absolute value of the correlation coefficient between S and $X_{t}$ for the dataset DATA3 was 0.74 (see Table 17), which can be confirmed by the viewpoint of Glennie and Lichti [15]. The calibration reference and calibration data collected in different coordinate systems will not significantly affect the calculation of the system error parameters. However, there was a high negative correlation between the ranging APs (*S* and *C*), namely −0.82, −0.81, and −0.79, respectively. The lower negative correlations between the ranging APs made the latter’s solution results more reliable. In these three calibration datasets, only points in planes B and J were with long pseudocalculated ranging measurements for calibration (see Figure 15). The calculated pseudoranging measurements from the points on these two planes ranged from about 35.3 m to 41.8 m for the dataset DATA1 (Table 7), from about 36.4 m to 44.2 m for the dataset DATA2 (Table 8), and from about 36.7 m to 45.5 m for the dataset DATA3 (Table 9). It demonstrated that the dataset with long calculated pseudoranging measurements for calibration would reduce the negative correlation between the ranging APs—the negative correlation of the dataset DATA3 is lower than those of the datasets DATA2 and DATA1. If there is a larger calibration site or suitable plan for scanning to collect the handheld LiDAR points to obtain more, longer calculated pseudoranging measurements for calibration, the negative correlation between the ranging APs would be reduced, and the solutions of *S* and *C* would be more reliable.

#### 3.5.4. The Analysis of Ranging Systematic Error Parameters

In this study, two ranging systematic parameters were estimated: *S* for the range scale factor and *C* for the rangefinder offset. Table 18 shows the estimated ranging systematic parameters and their precisions for the three datasets.

Table 19 indicates the correction for different distances. If the ranging measurement was 10 m, the correction was 1 cm for the dataset DATA1, 3 cm for the datasets DATA2 and DATA3; the ranging measurement was 30 m, the correction was 2 cm for the dataset DATA1, and 3 cm for the datasets DATA2 and DATA3; and the ranging measurement was 40 m, the correction was 2 cm for the dataset DATA1, and 4 cm for the datasets DATA2 and DATA3. This finding means that when scanning using a handheld LiDAR scanner, the closer to the object, the less the correction. However, even if it had been 2 meters, the correction would still have been 1–2 cm. Meanwhile, when using a handheld LiDAR scanner for precise surveying (e.g., cadastral surveying), these ranging system errors should be corrected to obtain more accurate results.

From Table 20, the difference of the correction on different dates on different distances (the datasets DATA1 and DATA2, DATA1 and DATA3) was about 1.2–2.0 cm, and the difference of the correction at different times on the same date at different distances (the datasets DATA2 and DATA3) was not larger than 2 mm. Further, it demonstrated the efficiency of the proposed calibration approach.

## 4. Conclusions

This study investigated the calibration results on different dates and at different times on the same date from the three datasets using the plane-based dynamic calibration method proposed by the previous study [23] for the GeoSLAM ZEB Horizon LiDAR scanner. Without considering the angle system error of the handheld LiDAR scanner, only two ranging system error parameters, including the range scale factor and the rangefinder offset, were calibrated.

After calibration, the distribution of residuals was more concentrated at 0, and the residual distribution was more in line with the normal distribution curve for the calibration data of the three datasets collected on different dates and at different times on the same date. For the three datasets, the average residuals were closer to 0, and the *a* posterior unit weight standard deviation became smaller, both of which were improved, compared to those without the additional ranging parameter in the adjustment. Therefore, the plane-based dynamic calibration method proposed by the previous study [23] used in this study could eliminate most of the GeoSLAM ZEB Horizon’s ranging system errors.

Based on the analysis of the RMSE results, the RMSE of all check planes was improved after correcting the ranging system error for the dataset DATA1. With up to 72.12% in one plane, an increase of about 2.4 cm was reached. The overall average improvement was 32.61%. For the dataset DATA2, with up to 61.08% in one plane, an increase of about 1.8 cm was reached. The overall average improvement was 28.44%. For the dataset DATA3, with up to 54.77% in one plane, an increase of about 1.4 cm was reached. The overall average improvement was 14.70%. Although the improvement of the dataset DATA3 was less than that of the dataset DATA2, the quality of the former’s point cloud was better than that of the dataset DATA2. The improvement is highly correlated with the quality of calibration data. The improvement of the RMSE of the check planes demonstrated again that the proposed calibration approach could effectively improve the overall point cloud accuracy of the GeoSLAM ZEB Horizon.

From the comparison of the RMSE difference of each check plane for the three calibration datasets, after calibration on different dates and at different times on the same date, the difference in the mean RMSE difference of the two sets of results was 0.0016 m, using data collected on different dates. On the other hand, the difference in the mean RMSE difference was only 0.0001 m when using calibration data on the same date but at different times to calibrate. There was no difference in the calibration results, showing the stable calibration on the different dates and at different times on the same date. This finding also demonstrated the efficiency of the proposed calibration approach and the calibration results during these two different dates.

From the investigation of the correlation between the additional ranging parameters *S* and *C*, the negative correlation between the ranging additional parameters *S* and *C* was −0.82, −0.81, and −0.79 for the three calibration datasets (DATA1, DATA2, and DATA3). The lower negative correlation between the ranging additional parameters makes the solution results of the ranging additional parameters *S* and *C* more reliable. Calibration data (DATA3) with longer calculated ranging measurements for calibration made the negative correlation between the ranging additional parameters, *S* and *C,* the least (−0.79). It demonstrated that the dataset with long calculated ranging measurements for calibration would reduce the negative correlation between the ranging APs; the negative correlation of the dataset DATA3 was lower than those of the datasets DATA2 and DATA1. Meanwhile, the calibration data of these three datasets employed about a 40–45 m longer calculated ranging measurements for calibration; the calibration data used in the previous study [23] employed only about 20 m calculated ranging measurements for calibration. Therefore, the negative correction in the previous study [23] is extremely high (−0.985), higher than those of the datasets used in this study. However, −0.79 was also high; if a larger calibration site or suitable plan for scanning is available for collecting the handheld LiDAR points with more, longer calculated ranging measurements for calibration in the future, it is believed that the negative correlation would be reduced, and the solutions of *S* and *C* would be more reliable.

The analysis of ranging systematic error parameters *S* and *C* concluded that when scanning by a handheld LiDAR scanner, the closer to the target, the lesser the correction. However, even if it were 2 meters, the correction would be 1–2 cm for ranging measurement. When using a handheld LiDAR scanner for precise surveying (e.g., cadastral surveying), these ranging system errors should be corrected to obtain more accurate results. The difference of the correction on different dates (the datasets DATA1 and DATA2, DATA1 and DATA3) at different distances was about 1–2 cm. On the other hand, the difference of the correction at different times on the same date (the datasets DATA2 and DATA3) at different distances was less than 1 mm. There was no difference in the calibration results, reiterating the stable calibration on the different dates and at different times on the same date in this study. Moreover, it demonstrated the efficiency of the proposed calibration approach.

## Figures and Tables

**Figure 1 sensors-22-00369-f001:**
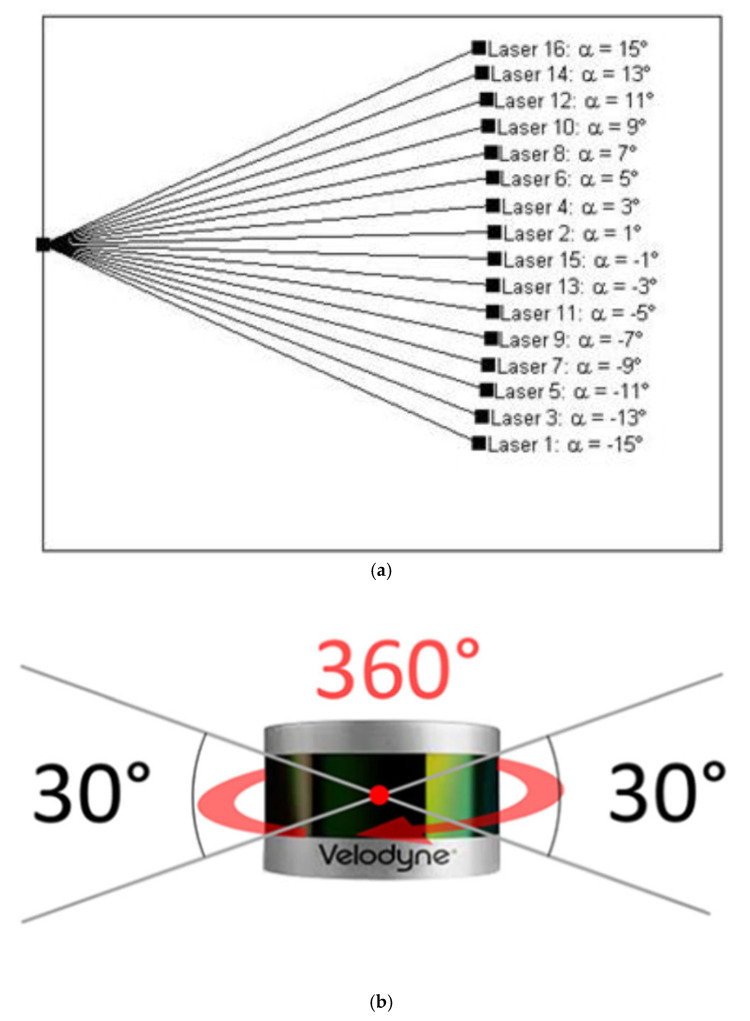
Velodyne VLP-16 sensor. (**a**) Internal laser positions [13]; (**b**) Sideview [14].

**Figure 2 sensors-22-00369-f002:**
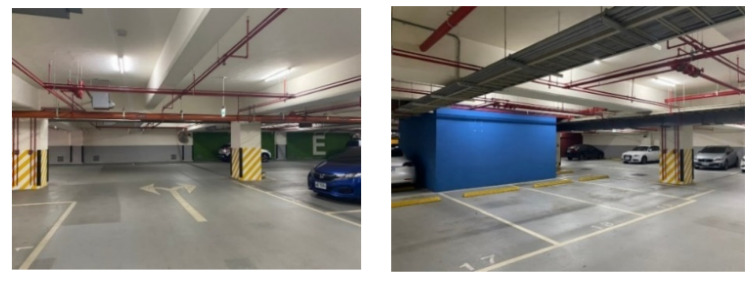
Calibration site.

**Figure 3 sensors-22-00369-f003:**
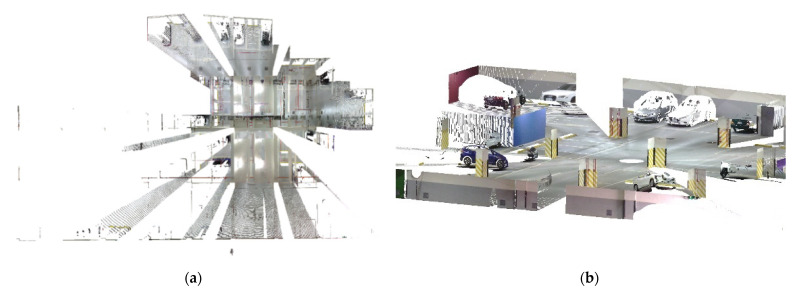
Top and side views of the collected points by a FARO Focus S350 in a single station (colored by RGB). (**a**) Top view; (**b**) Side view.

**Figure 4 sensors-22-00369-f004:**
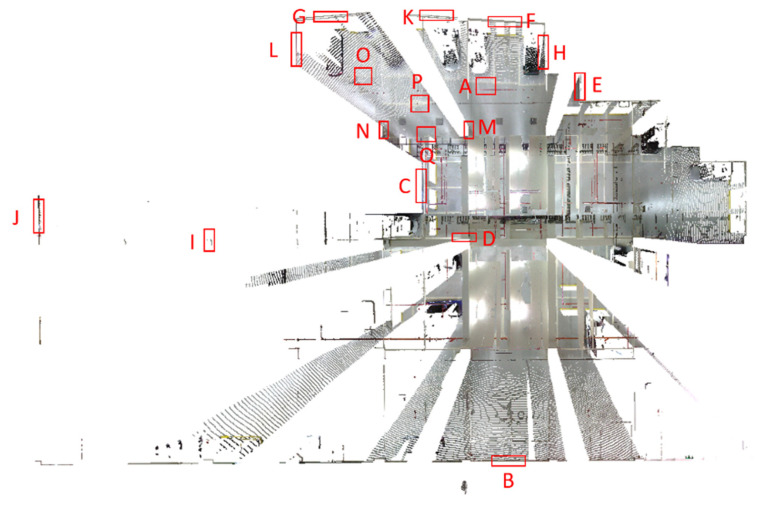
Locations of the selected 17 planes for calibration reference data.

**Figure 5 sensors-22-00369-f005:**
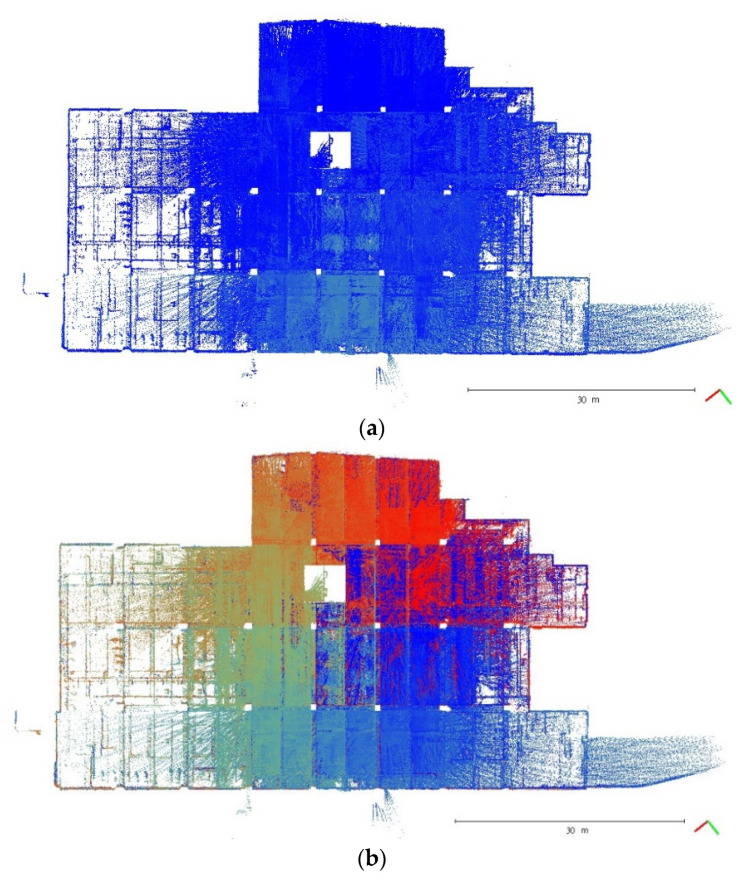
The top view of calibration dataset DATA1. (**a**) Point cloud colored by SLAM condition; (**b**) Point cloud colored by scanning time.

**Figure 6 sensors-22-00369-f006:**
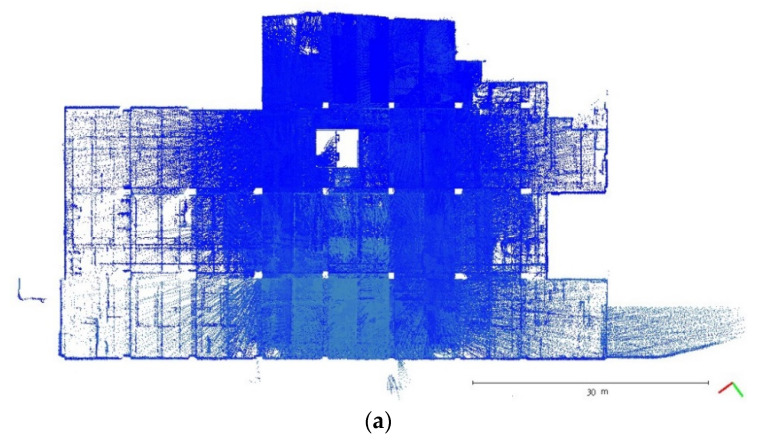
The top view of calibration dataset DATA2. (**a**) Point cloud colored by SLAM condition; (**b**) Point cloud colored by scanning time.

**Figure 7 sensors-22-00369-f007:**
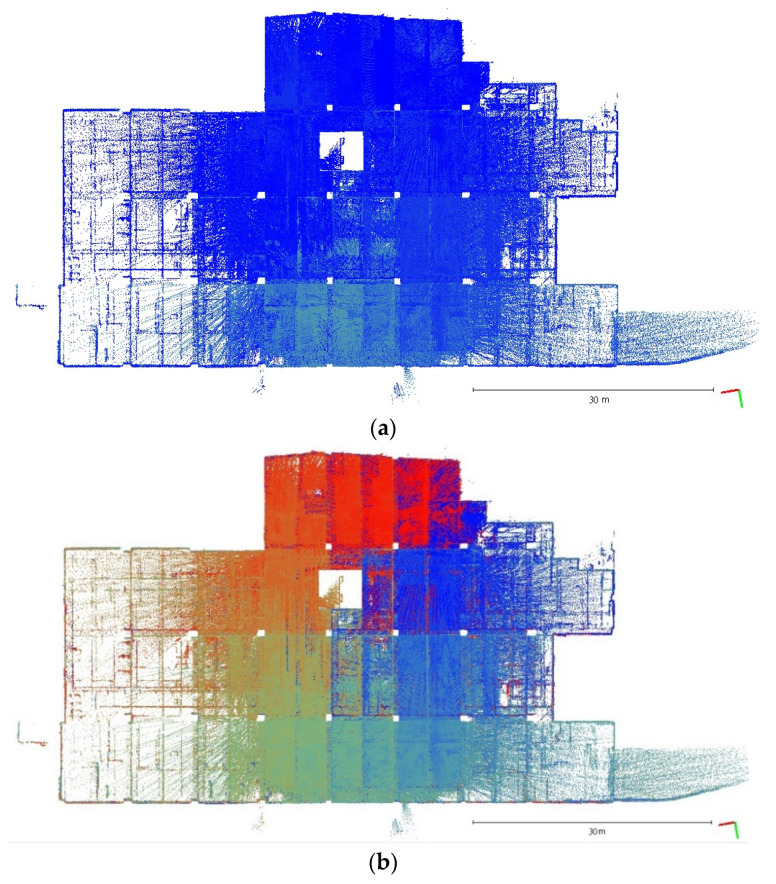
The top view of calibration dataset DATA3. (**a**) Point cloud colored by SLAM condition; (**b**) Point cloud colored by scanning time.

**Figure 8 sensors-22-00369-f008:**
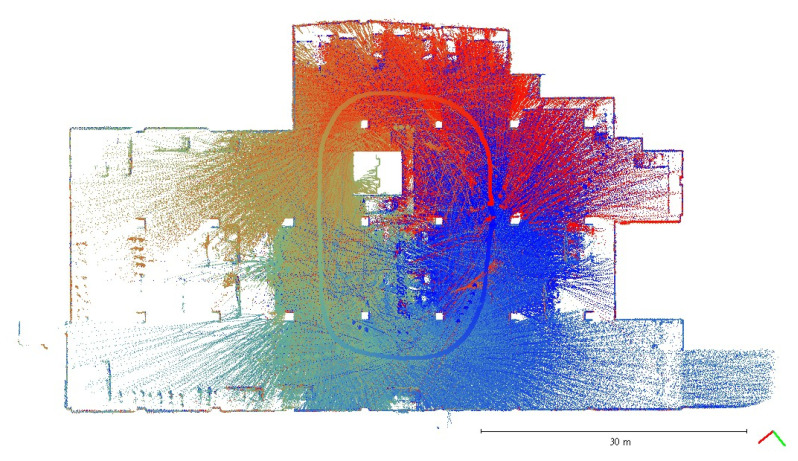
Point cloud colored by scanning time and the scanning trajectory of dataset DATA1.

**Figure 9 sensors-22-00369-f009:**
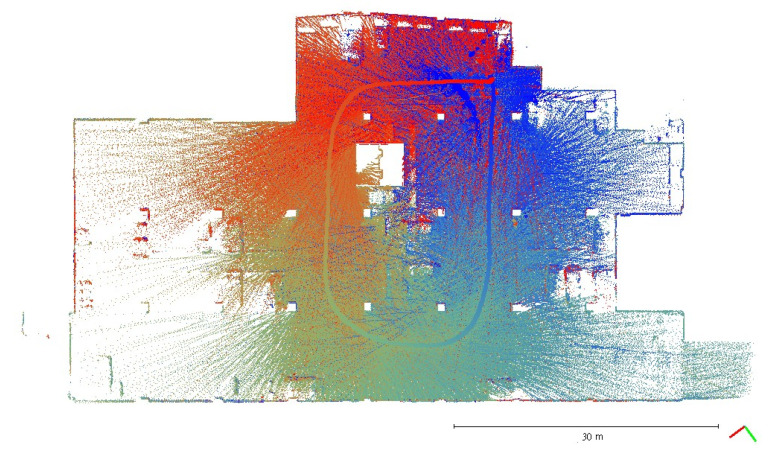
Point cloud colored by scanning time and the scanning trajectory of dataset DATA2.

**Figure 10 sensors-22-00369-f010:**
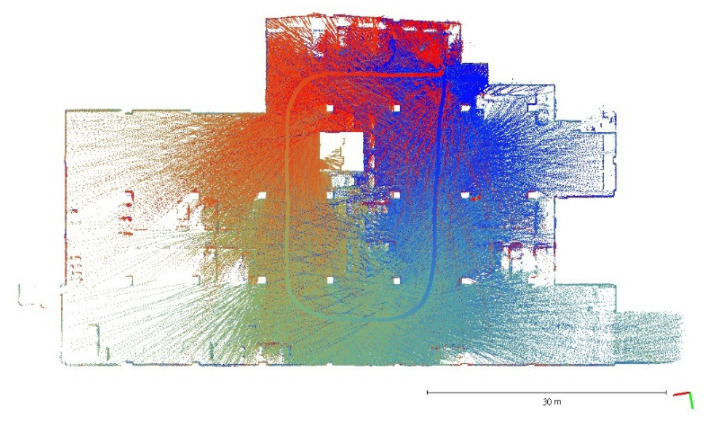
Point cloud colored by scanning time and the scanning trajectory of dataset DATA3.

**Figure 11 sensors-22-00369-f011:**
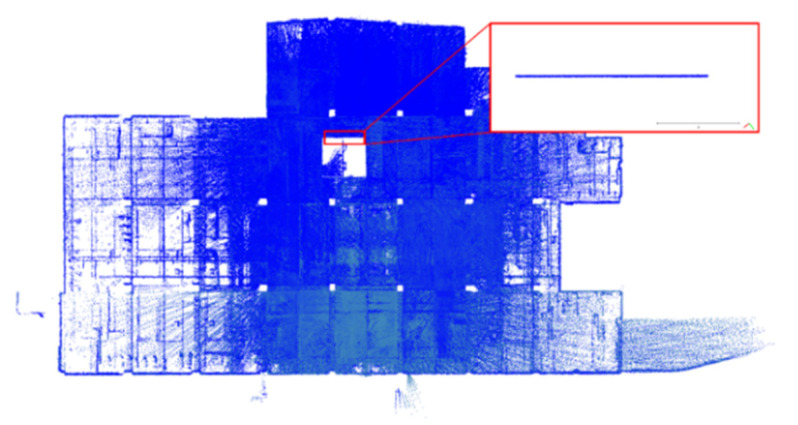
Planar point cloud location and point cloud data used for point cloud filtering analysis from dataset DATA1.

**Figure 12 sensors-22-00369-f012:**
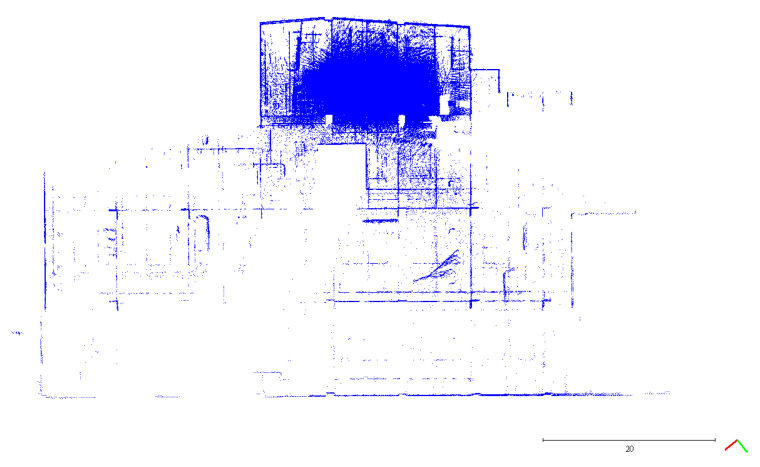
Handheld LiDAR point cloud after point cloud filtering for dataset DATA1.

**Figure 13 sensors-22-00369-f013:**
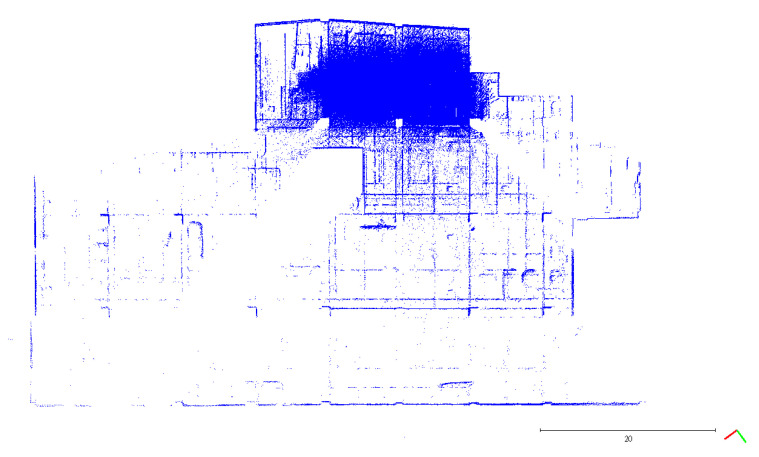
Handheld LiDAR point cloud after point cloud filtering for dataset DATA2.

**Figure 14 sensors-22-00369-f014:**
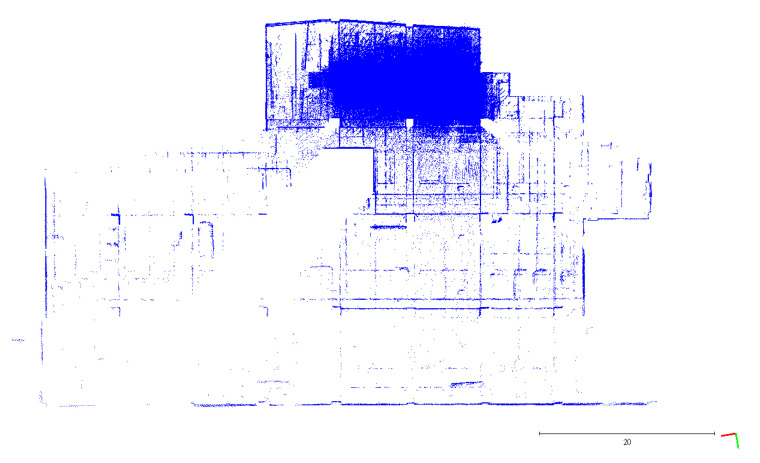
Handheld LiDAR point cloud after point cloud filtering for dataset DATA3.

**Figure 15 sensors-22-00369-f015:**
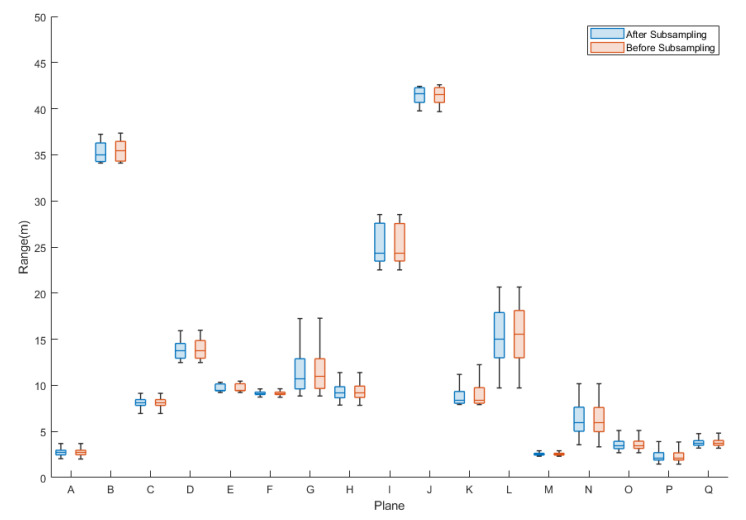
Box diagram of the calculated pseudoranging measurements of each plane, before and after subsampling and blunder removal by the RANSAC algorithm for dataset DATA1.

**Figure 16 sensors-22-00369-f016:**
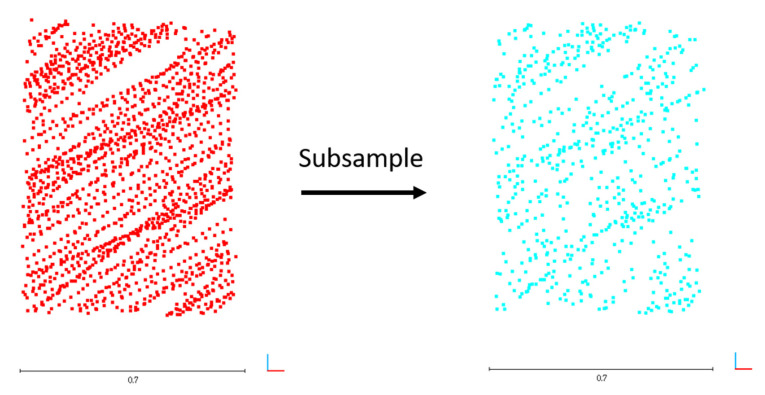
The illustration of subsampling results in plane *K*.

**Figure 17 sensors-22-00369-f017:**
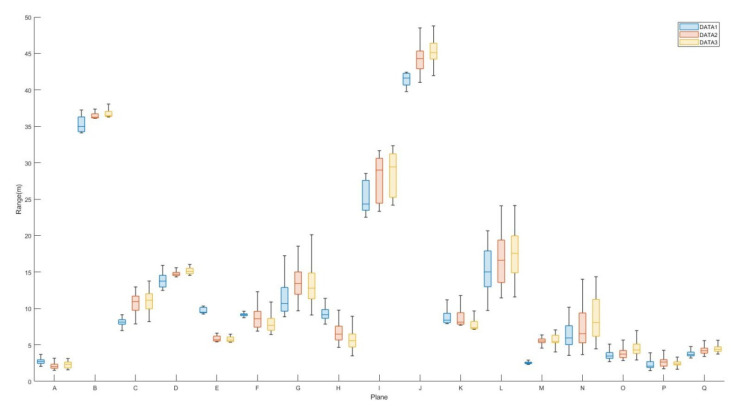
Box diagram of calculated pseudoranging measurements in each plane after subsampling and blunder removal by the RANSAC algorithm for datasets DATA1, DATA2, and DATA3.

**Figure 18 sensors-22-00369-f018:**
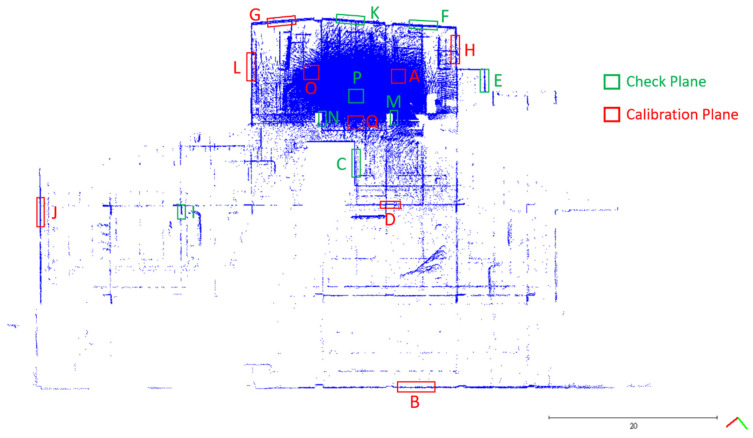
The locations of calibration planes and check planes.

**Figure 19 sensors-22-00369-f019:**
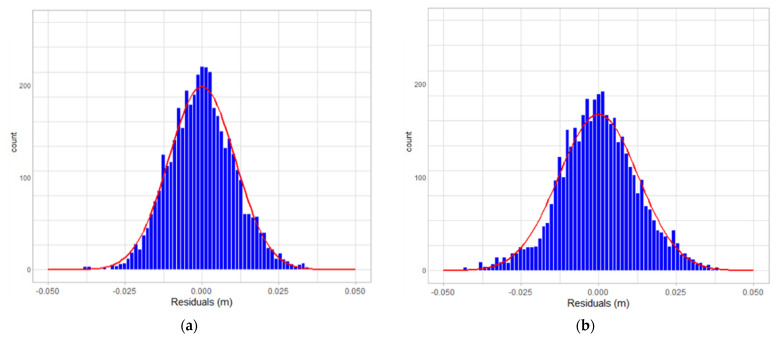
The residual distribution plots of dataset DATA1. (**a**) with Aps; (**b**) without Aps.

**Figure 20 sensors-22-00369-f020:**
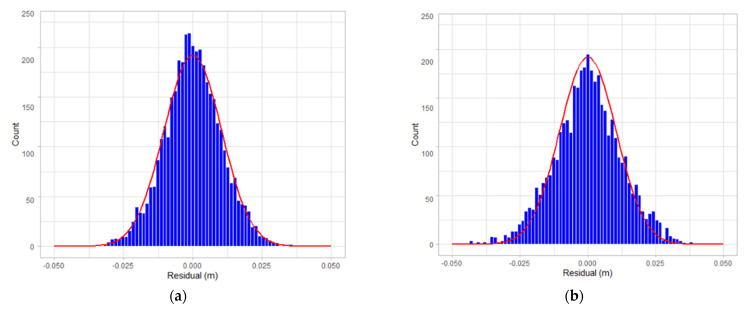
The residual distribution plots of dataset DATA2. (**a**) with Aps; (**b**) without Aps.

**Figure 21 sensors-22-00369-f021:**
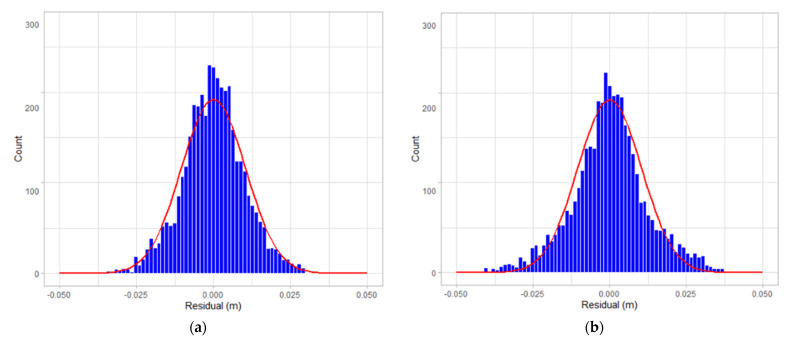
The residual distribution plots of dataset DATA3. (**a**) with Aps; (**b**) without Aps.

**Figure 22 sensors-22-00369-f022:**
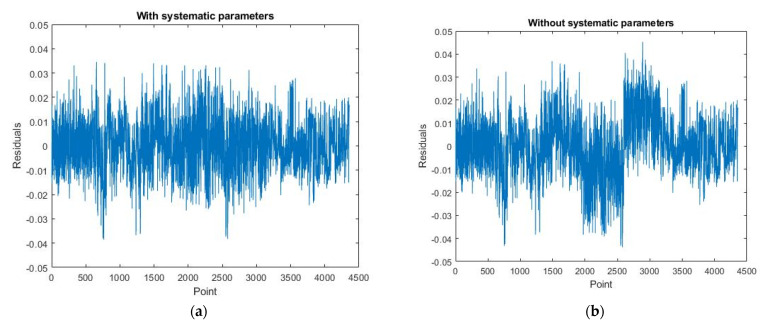
The residual scatter diagram of dataset DATA1. (**a**) with Aps; (**b**) without Aps.

**Figure 23 sensors-22-00369-f023:**
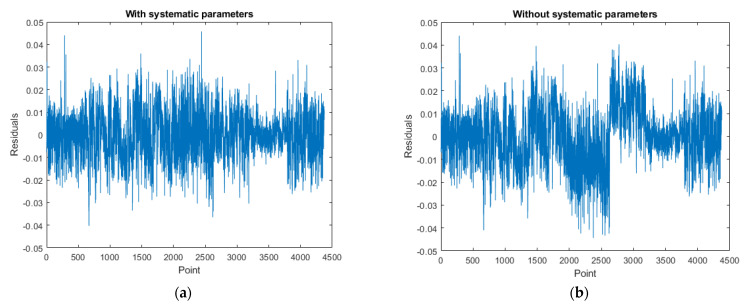
The residual scatter diagram of dataset DATA2. (**a**) with Aps; (**b**) without Aps.

**Figure 24 sensors-22-00369-f024:**
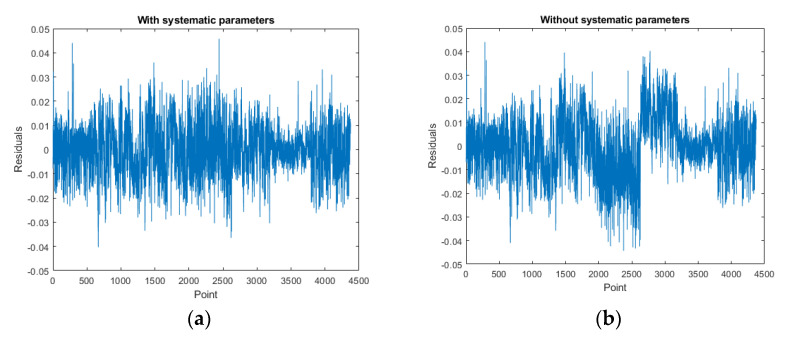
The residual scatter diagram of dataset DATA3. (**a**) with Aps; (**b**) without Aps.

**Figure 25 sensors-22-00369-f025:**
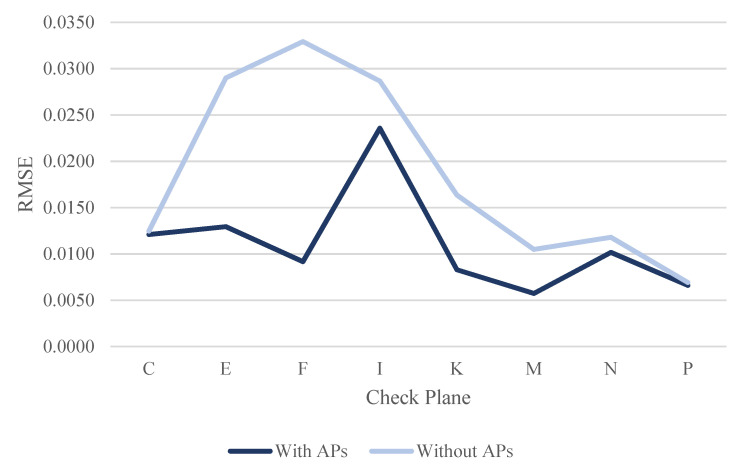
The line chart of RMSE for check planes in dataset DATA1.

**Figure 26 sensors-22-00369-f026:**
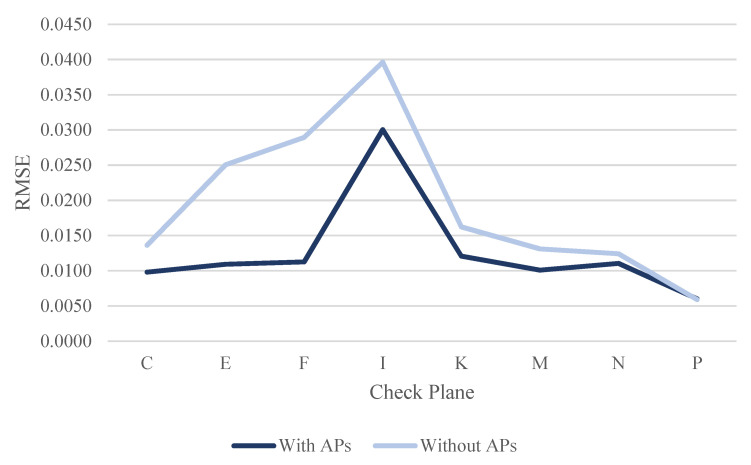
The line chart of RMSE for check planes in dataset DATA2.

**Figure 27 sensors-22-00369-f027:**
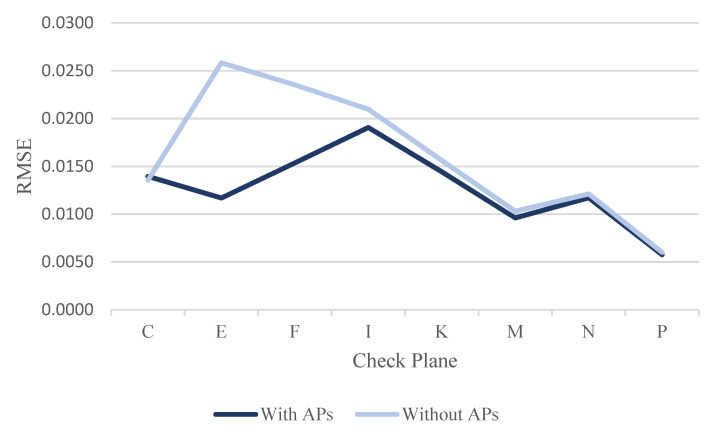
The line chart of RMSE for check planes in dataset Data3.

**Table 1 sensors-22-00369-t001:** Technical Information for FARO Focus S350 (https://echosurveying.com/3d-laser-scanner/faro-focus-s350-laser-scanner, accessed on 24 November 2021).

** 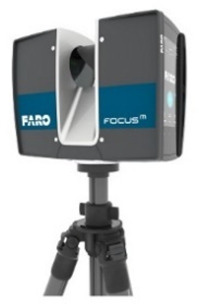 **	Range: 0.6—350 m
	High Dynamic Range (HDR) Photo Recording 2×/3×/5×
	Measurement Speed: up to 976,000 points/second
	Ranging Error: ±1 mm
	Sealed Design–Ingress Protection (IP) Rating Class 54
	On-site Compensation
	Accessory Bay
	Angular Accuracy: 19 arc sec for vertical/horizontal angles

**Table 2 sensors-22-00369-t002:** Technical specification for the GeoSLAM ZEB Horizon (https://microsolresources.com/wp-content/uploads/2019/06/GeoSLAM-Family-Brochure.pdf, accessed on 24 November 2021).

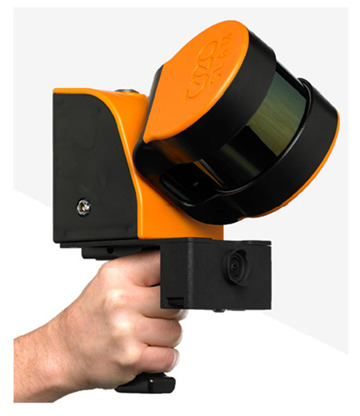	Technical specification	
	Handheld | Backpack | UAV Ready	
	Range	100 m
	Protection Class	IP54
	Scanner Weight	1.3 kg
	Points per Second	300,000
	Relative Accuracy	1–3 cm
	Raw Data File Size	100–200 MB a minute
	Processing	Point Processing
	Battery Life	3.5 h

**Table 3 sensors-22-00369-t003:** Fitting plane information.

Plane	a	b	c	d	Fitting RMSE (m)	DIP (°)
A	0.012	0.007	0.999	−44.416	0.0009	0
B	0.982	0.190	0.002	−23.319	0.0009	89
C	−0.196	0.981	−0.001	9.670	0.0008	89
D	0.982	0.190	0.001	−1.516	0.0006	89
E	−0.183	0.983	−0.003	−5.552	0.0008	89
F	0.993	0.121	0.002	19.513	0.0008	89
G	0.964	0.265	0.002	21.626	0.0006	89
H	−0.193	0.981	−0.002	−2.357	0.0007	89
I	−0.194	0.981	0.001	30.209	0.0006	89
J	−0.191	0.982	0.001	46.958	0.0010	89
K	0.994	0.112	0.003	19.384	0.0009	89
L	−0.189	0.982	−0.005	22.097	0.0006	89
M	−0.191	0.982	−0.001	5.064	0.0008	89
N	−0.187	0.982	−0.004	13.458	0.0005	89
O	−0.004	0.005	0.999	−44.549	0.0009	0
P	−0.003	−0.002	0.999	−44.667	0.0006	0
Q	0.002	0.003	0.999	−44.545	0.0010	0

**Table 4 sensors-22-00369-t004:** The relevant collected data for the three datasets.

Dataset Name	Date	Time Consumed	Point Numbers
DATA1	9:30 a.m. 26 January 2021	86.67 sec	15,734,365
DATA2	8:40 a.m. 11 May 2021	83.74 sec	15,020,154
DATA3	8:50 a.m. 11 May 2021	83.86 sec	15,000,136

**Table 5 sensors-22-00369-t005:** Analysis results of filtering conditions using a planar point cloud in dataset DATA1.

Filter Condition	Plane Fitting RMSE (m)	Number of Points	Filtering Points
No	0.0110	21,650	none
SLAM quality	0.0108	13,525	8125
incidence angle	0.0108	19,620	2024
SLAM quality and incidence angle	0.0106	12,969	8678

**Table 6 sensors-22-00369-t006:** The statistics of pseudoranging measurements in each plane for dataset DATA1.

Plane	No. of Points	Calculated Pseudoranging Measurement (m)			
		Minimum	Maximum	Median	Average
A	3024	2.012	4.166	2.714	2.745
**B**	**233**	**34.100**	**37.353**	**35.474**	**35.444**
C	660	6.953	9.137	8.137	8.138
D	613	12.471	15.977	13.775	13.962
**E**	**322**	**9.226**	**10.454**	**9.495**	**9.737**
F	862	8.713	13.018	9.125	9.347
G	847	8.846	18.200	10.954	11.536
H	1420	7.832	11.380	9.227	9.354
**I**	**80**	**22.515**	**28.518**	**24.317**	**24.985**
**J**	**91**	**39.675**	**45.099**	**41.515**	**41.833**
K	1595	7.916	12.618	8.396	9.029
L	864	9.722	20.655	15.538	15.484
M	1110	2.310	2.909	2.542	2.546
N	1417	3.333	10.170	5.961	6.335
O	818	2.695	12.473	3.485	3.722
P	2119	1.456	7.881	2.082	2.342
Q	736	3.186	7.054	3.698	3.873

**Table 7 sensors-22-00369-t007:** The statistics of pseudoranging measurement in each plane after subsampling and blunder removal by the RANSAC algorithm for dataset DATA1.

Plane	No. of Points	Calculated Pseudoranging Measurement (m)			
		Minimum	Maximum	Median	Average
A	586	2.043	4.043	2.715	2.738
**B**	**211**	**34.100**	**37.230**	**34.982**	**35.313**
C	595	6.953	9.137	8.146	8.138
D	567	12.471	15.919	13.768	13.923
**E**	**297**	**9.226**	**10.328**	**9.474**	**9.691**
F	593	8.735	12.964	9.125	9.337
G	579	8.846	18.200	10.696	11.449
H	581	7.860	11.380	9.161	9.317
**I**	**58**	**22.515**	**28.518**	**24.350**	**25.192**
**J**	**72**	**39.757**	**45.099**	**41.605**	**41.836**
K	592	7.919	12.618	8.357	8.949
L	569	9.722	20.655	14.993	15.391
M	599	2.310	2.906	2.536	2.542
N	562	3.555	10.170	5.987	6.391
O	600	2.695	12.473	3.493	3.706
P	599	1.466	7.295	2.088	2.348
Q	593	3.199	7.054	3.694	3.866

**Table 8 sensors-22-00369-t008:** The statistics of calculated ranging measurement in each plane after subsampling and blunder removal by the RANSAC algorithm for dataset DATA2.

Plane	No. of Points	Calculated Pseudoranging Measurement (m)			
		Minimum	Maximum	Median	Average
A	591	1.505	6.057	1.981	2.158
B	236	36.083	37.362	36.271	**36.433**
C	592	7.878	12.947	10.924	10.748
D	576	14.318	15.713	14.686	14.775
E	559	5.416	6.609	5.702	5.827
F	597	6.884	12.306	8.553	8.559
G	574	9.663	21.458	13.432	13.713
H	580	4.657	11.851	6.479	6.799
I	62	23.331	31.650	28.967	27.930
J	69	41.010	48.504	44.266	**44.169**
K	575	7.695	14.864	8.149	8.893
L	560	11.449	24.081	16.594	16.654
M	576	4.004	6.354	5.520	5.478
N	590	3.660	14.014	6.524	7.445
O	598	2.845	16.973	3.717	4.016
P	600	1.725	8.051	2.675	2.715
Q	589	3.408	9.909	4.167	4.327

**Table 9 sensors-22-00369-t009:** The statistics of calculated ranging measurement in each plane after subsampling and blunder removal by the RANSAC algorithm for dataset DATA3.

Plane	No. of Points	Calculated Pseudoranging Measurement (m)			
		Minimum	Maximum	Median	Average
A	594	1.573	6.283	2.316	2.524
B	277	36.278	38.089	36.456	**36.736**
C	580	8.192	13.758	11.132	11.069
D	539	14.543	16.038	15.071	15.127
E	582	5.371	6.472	5.649	5.744
F	560	6.418	12.203	7.655	8.035
G	594	9.091	21.843	12.810	13.390
H	559	3.494	8.933	5.578	5.661
I	57	24.172	32.324	29.450	28.643
J	108	41.964	48.780	45.117	**45.451**
K	585	7.129	15.136	7.368	8.014
L	559	11.568	24.126	17.556	17.679
M	600	4.038	7.059	5.483	5.701
N	583	4.454	14.347	8.052	8.629
O	596	2.931	15.268	4.287	4.743
P	600	1.656	5.764	2.423	2.555
Q	556	3.738	10.547	4.384	4.643

**Table 10 sensors-22-00369-t010:** Average residual and unit weight standard deviation for the three datasets using least-squares adjustment, with and without ranging APs.

Dataset	Adjustment with Ranging APs	Average Residual $\bar{v}$	A Posterior Unit Weight Standard Deviation $\sigma_{0}$
DATA1	**Yes**	**−0.000014845 m**	**±0.01077 m**
	No	−0.000045195 m	±0.01278 m
DATA2	**Yes**	**−0.000010804 m**	**±0.01040 m**
	No	−0.000044651 m	±0.01240 m
DATA3	**Yes**	**−0.000016866 m**	**±0.01012 m**
	No	−0.000061457 m	±0.01240 m

**Table 11 sensors-22-00369-t011:** RMSE of each check plane for dataset DATA1.

Check Plane	$RMSE_{withAPs}\;(m)$	$RMSE_{withoutAPs}\;(m)$	Difference (m)	Improvement (%)
**C**	0.0121	0.0125	0.0004	2.98%
**E**	**0.0129**	**0.0290**	**0.0161**	**55.35%**
** F **	** 0.0092 **	** 0.0329 **	**0.0237**	** 72.12% **
I	*0.0236*	0.0287	0.0051	17.75%
K	0.0083	0.0164	0.0081	49.25%
M	0.0057	0.0105	0.0048	45.27%
N	0.0102	0.0118	0.0016	13.82%
P	0.0066	0.0069	0.0003	4.32%
**Mean**	0.0111	0.0186	0.0075	**32.61%**

**Table 12 sensors-22-00369-t012:** RMSE of each check plane for dataset DATA2.

Check Plane	$RMSE_{withAPs}\;(m)$	$RMSE_{withoutAPs}\;(m)$	Difference (m)	Improvement (%)
C	0.0098	0.0136	0.0038	27.98%
**E**	**0.0109**	**0.0251**	**0.0142**	**56.42%**
** F **	** 0.0113 **	** 0.0289 **	**0.0176**	** 61.08% **
I	*0.0300*	0.0396	0.0096	24.19%
K	0.0121	0.0162	0.0041	25.59%
M	0.0101	0.0131	0.003	23.05%
N	0.0110	0.0124	0.0014	11.03%
*P*	*0.0060*	*0.0059*	−0.0001	*−1.86%*
**Mean**	0.0127	0.0194	0.0067	**28.44%**

**Table 13 sensors-22-00369-t013:** RMSE of each check plane for dataset DATA3.

Check Plane	$RMSE_{withAPs}\;(m)$	$RMSE_{withoutAPs}\;(m)$	Difference (m)	Improvement (%)
*C*	*0.0139*	*0.0136*	−0.0003	*−2.90%*
**E**	**0.0117**	**0.0258**	**0.0141**	**54.77%**
F	0.0154	0.0235	0.0081	34.68%
I	** *0.0191* **	0.0210	0.0019	9.13%
K	0.0144	0.0156	0.0012	7.65%
M	0.0096	0.0103	0.0007	6.77%
N	0.0117	0.0121	0.0004	3.33%
P	0.0058	0.0060	0.0002	4.19%
**Mean**	0.0127	0.0160	0.0033	14.70%

**Table 14 sensors-22-00369-t014:** The comparison of RMSE difference of each check plane for the three datasets after calibration.

	Different Date		Different Time
Plane	RMSE difference (m) DATA2-DATA1	RMSE difference (m) DATA3-DATA1	RMSE difference (m) DATA3-DATA2
	**Collected Time:** **DATA1: 9:30 a.m. 25 January 2021; DATA2: 8:40 a.m. 11 May 2021; DATA3: 8:50 a.m. 11 May 2021**		
*C*	−0.0023	0.0018	0.0041
E	−0.0020	−0.0012	0.0008
F	0.0021	0.0062	0.0041
I	0.0064	−0.0045	−0.0109
K	0.0038	0.0061	0.0023
M	0.0044	0.0039	−0.0005
N	0.0008	0.0015	0.0007
P	−0.0006	−0.0008	−0.0002
**Mean**	**0.0016**	**0.0016**	**0.0001**
Max(Abs)	0.0064	0.0062	0.0109
Min(Abs)	0.0006	0.0008	0.0002

**Table 15 sensors-22-00369-t015:** The matrix of correlation coefficients of the unknowns for dataset DATA1.

	*S*	*C*	$X_{t}$	$Y_{t}$	$Z_{t}$	φ	ω	κ
*S*	1	−0.82	−0.68	0.65	−0.04	−0.06	0.12	−0.61
*C*	−0.82	1	0.30	−0.28	0.08	0.06	−0.14	0.23
$X_{t}$	−0.68	0.30	1	−0.91	0.10	−0.08	0.07	0.94
$Y_{t}$	0.65	−0.28	−0.91	1	0.03	−0.05	−0.03	−0.94
$Z_{t}$	−0.04	0.08	0.10	0.03	1	−0.97	0.54	0.00
φ	−0.06	0.06	−0.08	−0.05	−0.97	1	−0.59	0.01
ω	0.12	−0.14	0.07	−0.03	0.54	−0.59	1	−0.02
κ	−0.61	0.23	0.94	−0.94	0.00	0.01	−0.02	1

**Table 16 sensors-22-00369-t016:** The matrix of correlation coefficients of the unknowns for dataset DATA2.

	*S*	*C*	$X_{t}$	$Y_{t}$	$Z_{t}$	φ	ω	κ
*S*	1	−0.81	−0.70	0.55	−0.41	−0.18	0.31	−0.68
*C*	−0.81	1	0.30	−0.32	0.52	0.21	−0.36	0.31
$X_{t}$	−0.70	0.30	1	−0.56	0.08	−0.05	0.00	0.92
$Y_{t}$	0.55	−0.32	−0.56	1	0.04	−0.23	0.04	−0.56
$Z_{t}$	−0.41	0.52	0.08	0.04	1	0.07	−0.74	0.16
φ	−0.18	0.21	−0.05	−0.23	0.07	1	−0.63	0.12
ω	0.31	−0.36	0.00	0.04	−0.74	−0.63	1	−0.17
κ	−0.68	0.31	0.92	−0.56	0.16	0.12	−0.17	1

**Table 17 sensors-22-00369-t017:** The matrix of correlation coefficients of the unknowns for dataset DATA3.

	*S*	*C*	$X_{t}$	$Y_{t}$	$Z_{t}$	φ	ω	κ
*S*	1	−0.79	−0.74	0.58	−0.35	−0.19	0.23	−0.71
*C*	−0.79	1	0.31	−0.31	0.46	0.22	−0.27	0.31
$\;X_{t}$	−0.74	0.31	1	−0.57	0.07	−0.06	−0.03	0.92
$\;Y_{t}$	0.58	−0.31	−0.57	1	0.07	−0.15	−0.13	−0.55
$\;Z_{t}$	−0.35	0.46	0.07	0.07	1	0.28	−0.90	0.14
φ	−0.19	0.22	−0.06	−0.15	0.28	1	−0.37	0.11
ω	0.23	−0.27	−0.03	−0.13	−0.90	−0.37	1	−0.12
κ	−0.71	0.31	0.92	−0.55	0.14	0.11	−0.12	1

**Table 18 sensors-22-00369-t018:** Estimated ranging systematic parameters for the three datasets.

Dataset	DATA1		DATA2		DATA3	
Ranging APs	*S*	*C*	*S*	*C*	*S*	*C*
Value	0.99964	−0.00884	0.99983	−0.01171	0.99992	−0.01395
Standard deviation	0.00004	0.00055	0.00004	0.00055	0.00004	0.00052

**Table 19 sensors-22-00369-t019:** The different distance values after correction (unit: m).

Distance	1	2	5	10	20	30	40	50
DATA1	0.99	1.99	4.99	9.99	19.98	29.98	39.98	49.97
DATA2	0.98	1.98	4.98	9.97	19.97	29.96	39.96	49.95
DATA3	0.98	1.98	4.98	9.97	19.97	29.96	39.96	49.96

**Table 20 sensors-22-00369-t020:** The difference of correction for the three datasets at different distances (unit: m).

Distance Difference	1	2	5	10	20	30	40	50
DATA1-DATA2	0.012	0.012	0.013	0.013	0.015	0.017	0.018	0.020
DATA1-DATA3	0.014	0.014	0.014	0.015	0.016	0.016	0.017	0.018
**DATA2-DATA3**	**0.002**	**0.002**	**0.002**	**0.001**	**0.000**	**0.000**	**−0.001**	**−0.002**

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
