# Peer review of "An Investigation on a Plane-Based Dynamic Calibration Method for the Handheld LiDAR Scanner"

_sensors, 2022, doi:10.3390/s22010369_

Round 1
Reviewer 1 Report
- Line 7: It is not common to use references in the abstract. Please remove.
- Line 39: Figure 1 is not clear enough and too brief. Please add a LiDAR side view to describe in details.
- Line 96: 'RMSE' first appears in line 13 but is explained in line 96. Acronyms must be defined/explained before first use.
- Line 188: It is mentioned that the extracted plane features were classified into the calibration planes and check planes. Please describe how to classify with formulas and point cloud figures.
- Line 247: Please adjust the format of formulas according to template.
- The number of reference is not enough.
- Add more captions to figures.
Author Response
Please see the upload file. Thanks.

Reviewer 2 Report
Dear author, I deal with the distance-measuring technique and laser scanning on a daily basis. In my opinion, calibrating a laser scanner should be the manufacturer's domain because only he has access to the appropriate analog and digital components of the scanner that enable its calibration. Manufacturers of higher-end scanners define the time intervals after which calibration is required. Of course, it is possible to perform this task by the user, but it always requires a lot of time and creating conditions for the implementation of such a task, but the question is for how long it will be enought. The situation is even more difficult if you try to get higher accuracies than guaranteed by the manufacturer. I believe that the publication may be interesting if the following remarks are met:
- The work [1], which is frequently cited and developed in this article, cannot be accessed online.
- The work is very extensive with many results, while reading it is difficult to understand what the author wants to show
- On line 84 the term "eccentricity error" is used after [17] and [18], I found no explanation in these works
- Why are Figures 8, 9 and 10 shown?
- Why are tables 10, 11 and 12 listed what is the purpose?
After reading this work, I could not understand what is "A plane-based dynamic calibration method"
Author Response
Please see the upload file. Thanks.

Reviewer 3 Report
This paper describes a calibration technique for finding the range scale factor and rangefinder offset for a handheld lidar scanner. The calibration was computed and verified using a high-quality lidar scanner dataset as the reference dataset. The targets for calibration were from a parking garage with several flat planes in the scanned point cloud. Both calibration planes and “check” planes were used. The planes were manually selected and the method for selecting points on these planes was described. The desired calibration parameters were estimated along with pose parameters needed to rotate the handheld data to the reference scanner pose. A weakness of the paper is that it would be nice to have the calibration parameters for each of the lidar Tx-Rx channels, but for the scanner used it appears that information is not available for each lidar point. Additionally, the study is for one scanner model. Verification of the method for other models would be helpful.
The proposed method is verified using residual error analysis, RMSE error on the check planes constructed from the data, and parameter correlation matrices. Analysis is presented for datasets on the same day at different times and different days.
The paper is well written and easy to follow.
I would recommend that the paper be accepted after minor revision including a few grammar and typo errors.
Author Response
Please see the upload file. Thanks.

Round 2
Reviewer 2 Report
Dear author, by saying that I don't understand what "......................................." is, I meant that the work is too detailed in many places and therefore distracts from the main plot, I still think that there is no need to quote entire tables 7 , 8, 9, the fragments as in the answer to comment 5 are enough, but the choice is Yours.
I found a typo "multi-bean" on line 85.
Congratulations on your interesting article.
Author Response
I agree your viewpoint.” There is no need to quote entire tables 7, 8, 9, the fragments as in the answer to comment 5 are enough”, therefore, Tables 7,8, and 9 in this reversion has been deleted. Meanwhile, the relevant table no. was also revised. The typo error “RANSC” has been revised and thanks for your congratulations. Also see the response in the submitted word file.
